

# A recombinase polymerase amplification-lateral flow dipstick assay for rapid detection of the quarantine citrus pathogen in China, *Phytophthora hibernalis*

Tingting Dai[1], Tao Hu[1], Xiao Yang[2,3], Danyu Shen[4], Binbin Jiao[5], Wen Tian[6] and Yue Xu[1]

[1] Co-Innovation Center for the Sustainable Forestry in Southern China, Nanjing Forestry University, Nanjing, Jiangsu, China
[2] United States Department of Agriculture, Agricultural Research Service, Foreign Disease-Weed Science Research Unit, Fort Detrick, MD, USA
[3] Oak Ridge Institute for Science and Education, ARS Research Participation Program, Oak Ridge, TN, USA
[4] Department of Plant Pathology, Nanjing Agricultural University, Nanjing, China
[5] Technical Center for Animal, Plant and Food Inspection and Quarantine of Shanghai Customs, Shanghai, China
[6] Jiangyin Customs House, Jiangyin, China

Corresponding author
Tingting Dai,
13770647123@163.com

## ABSTRACT

*Phytophthora hibernalis*, the causal agent of brown rot of citrus fruit, is an important worldwide pathogen and a quarantine pest in China. Current diagnosis of the disease relies on disease symptoms, pathogen isolation and identification by DNA sequencing. However, symptoms caused by *P. hibernalis* can be confused with those by other *Phytophthora* and fungal species. Moreover, pathogen isolation, PCR amplification and sequencing are time-consuming. In this study, a rapid assay including 20-min recombinase polymerase amplification targeting the *Ypt1* gene and 5-min visualization using lateral flow dipsticks was developed for detecting *P. hibernalis*. This assay was able to detect 0.2 ng of *P. hibernalis* genomic DNA in a 50-μL reaction system. It was specific to *P. hibernalis* without detection of other tested species including *P. citrophthora*, *P. nicotianae*, *P. palmivora* and *P. syringae*, four other important citrus pathogens. Using this assay, *P. hibernalis* was also detected from artificially inoculated orange fruits. Results in this study indicated that this assay has the potential application to detect *P. hibernalis* at diagnostic laboratories and plant quarantine departments of customs, especially under time- and resource-limited conditions.

## INTRODUCTION

*Phytophthora hibernalis* is an important pathogen causing brown rot (*Erwin & Ribeiro, 1996*) and gummosis (*Graham & Menge, 2000*) of citrus. It also occasionally causes diseases of other crops such as tomato and apple (*Erwin & Ribeiro, 1996*). Due to its

potential impact on the horticultural industry, this exotic pathogen has been listed as a quarantine pest in China since 2007 (*Li et al., 2019*). In 2015, *P. hibernalis* was detected in citrus fruit shipments from Tulare County, CA, USA to Shanghai, China, leading to a suspension of importing CA citrus products to China from February to November 2015 (*Hao et al., 2018*; *Jiao et al., 2017*). Such restrictions caused by this quarantine pathogen have caused significant losses to Californian citrus farmers and international trade and shipping companies in both export and import countries (*Adaskaveg & Förster, 2014*; *Hao et al., 2018*; *Jiao et al., 2017*).

Successful quarantine measures are established on accurate and timely detection of pathogens. Traditionally, diagnoses include pathogen isolation and identification based on the morphological and molecular features. This process could be time-consuming. Therefore, it hardly meets the need of rapid detection at customs. With the advent of molecular technology, other faster and more cost-effective techniques including isothermal diagnostic assays have become the mainstream in rapid detection of many *Phytophthora* species (*Miles, Martin & Coffey, 2015*) such as *P. infestans* (*Hansen et al., 2016*; *Khan et al., 2017*; *Si Ammour et al., 2017*), *P. sojae* (*Dai et al., 2012*; *Dai, Zheng & Wu, 2015*; *Rojas et al., 2017*), *P. nicotianae* (*Li et al., 2015*), *P. cinnamomi* (*Dai et al., in press*). A loop-mediated isothermal amplification (LAMP) assay for detecting *P. hibernalis* was developed by *Li et al. (2019)*.

The recombinase polymerase amplification (RPA) is an emerging isothermal nucleic acid amplification technique for detecting plant pathogens. The RPA enzyme mixture conducts exponential amplification of the target region within the DNA template. This was accomplished by a phage-derived recombinase UvsX, which aggregates with two oligonucleotide primers to form a nucleoprotein filament and scan for single homologous target sequence of the DNA template (*Piepenburg et al., 2006*). A single-strand DNA-binding protein binds to the filament. A strand displacing DNA polymerase then amplifies the template (*Daher et al., 2016*; *James & Macdonald, 2015*; *Piepenburg et al., 2006*). RPA has several advantages compared to conventional PCR-based methods and LAMP assays (*Piepenburg et al., 2006*). First, this technique amplifies DNA at constant and relatively low temperatures (optimal range, 37–42 °C), therefore, eliminating the requirement of thermal cyclers. Second, each RPA assay requires only a pair of oligonucleotide primers, while the LAMP technique requires four to six primers to synthesize various sizes of DNA amplicons (*Notomi et al., 2000*). Third, RPA amplicons can be visualized in real-time using lateral flow dipsticks (LFDs). The RPA–LFD assay requires an oligonucleotide probe in addition to the primer pair. The probe is conjugated with a fluorescein amidite (FAM) and a C3 spacer on its 5′ and 3′ ends, respectively, and a base analog tetrahydrofuran (THF) inserted as an internal base. Subsequently, the amplicons generated by the RPA primers and probe are visualized in the "sandwich" assay (*Ghosh et al., 2018*; *Hou et al., 2018*).

In this study, a novel RPA assay targeting the *ras*-related *Ypt1* gene of *P. hibernalis* was developed. Specificity of this assay was evaluated by testing against an array of oomycete and fungal species. Moreover, sensitivity of this assay was compared with a conventional PCR-based method.

## MATERIALS AND METHODS

### RPA primers and probe design

Sequences of the *Ypt1* gene of *P. hibernalis* (GenBank Accession No. KJ755160) and other *Phytophthora* species were obtained from GenBank (*Benson et al., 2018*). Alignment of these sequences (Fig. S1) was carried out using Clustal W (*Larkin et al., 2007*). Six pairs of primers (Table S1) targeting the polymorphic regions of the *Ypt1* gene were designed for *P. hibernalis* according to the TwistAmp® DNA Amplification Kits Assay Design Manual (https://www.twistdx.co.uk/docs/default-source/RPA-assay-design/twistamp-assay-design-manual-v2-5.pdf?sfvrsn=29). Their specificity to *P. hibernalis* was preliminarily evaluated using a collection of *Phytophthora* species (Table 1) in conventional PCR. A primer pair, including a forward primer PhRPA-F and a reverse primer PhRPA-R was found exclusively and consistently amplified the *Ypt1* gene of *P. hibernalis* isolates. The reverse primer was then adjusted by conjugating biotin at its 5′-end (Table 2). A probe PhProbe (Table 2) was designed according to the TwistAmp® DNA Amplification Kits Assay Design Manual. This set of primers and probe was tested in RPA and subsequently used for optimizing the reaction condition and evaluating assay specificity and sensitivity. Primers and probe (Table 2) were synthesized by Nanjing GenScript Co. Ltd. (Nanjing, China).

### Isolate selection

Isolates of oomycete and fungal species used in this study are listed in Table 1. *P. hibernalis* isolate 9099 was recovered from an imported pomelo fruit (*Citrus grandis*) at Shanghai Export and Import Inspection and Quarantine Bureau (*Jiao et al., 2017*). All isolates were part of collections at Department of Plant Pathology at Nanjing Agricultural University and Department of Forest Protection at Nanjing Forestry University in Nanjing, China.

### Culture conditions and DNA extraction

Isolates of *Phytophthora* species were cultured in 10% clarified V8 juice agar (cV8A) at 18 to 25 °C in the dark. Mycelia of oomycete and fungal isolates were produced by growing individual isolates in 20% clarified V8 juice and potato dextrose broth, respectively, at 18–25 °C for 3–5 days. Mycelia were harvested by filtration and then frozen at −20 °C. Genomic DNAs (gDNA) were extracted using a DNAsecure Plant Kit (Tiangen Biotech, Beijing, China). DNA concentrations were measured by a NanoDrop 2000c spectrophotometer (NanoDrop Technologies, Thermo Fisher Scientific, Wilmington, DE, USA). Nuclease-free water (nfH$_2$O; Thermo Fisher Scientific, Wilmington, DE, USA) was used to dilute gDNA extractions as needed. DNA samples were stored at −20 °C until use.

### RPA–LFD assay

RPA amplifications were performed in 50-μL reactions according to the quick guide of the TwistAmp® nfo Kit (TwistDx Ltd., Cambridge, UK). After mixing 2.1 μL of each of PhRPA-F and PhRPA-R primers (10 μM), 0.6 μL of PhProbe probe (10 μM), 29.5 μL of rehydration buffer (supplied in the kit) and 11.2 μL of nfH$_2$O, the reagent mixture of

**Table 1 List of isolates used in the study and their detection results in the recombinase polymerase amplification-lateral flow dipstick (RPA–LFD) assay.**

| (Sub)clade[a] | Species | Isolate | Host/substrate | Location[b] | RPA–LFD[c] |
|---|---|---|---|---|---|
| 8c | *Phytophthora hibernalis* | 9099 | *Citrus grandis* | SH, China | + |
| | *P. hibernalis* | CBS 270.31 | *Citrus sinensis* | Setúbal, Portugal | + |
| | *P. lateralis* | CBS 168.42 | *Chamaecyparis lawsoniana* | OR, USA | – |
| | *P. ramorum* | EU1 2275 | *Laurus nobilis* | UK | – |
| 8a | *P. cryptogea* | Pcr1 | *Gerbera jamesonii* | JS, China | – |
| | *P. drechsleri* | CBS 292.35 | *Beta vulgaris* var. *altissima* | CA, USA | – |
| | *P. erythroseptica* | CBS 129.23 | *Solanum tuberosum* | Ireland | – |
| | *P. medicaginis* | ATCC 44390 | *Medicago sativa* | CA, USA | – |
| 8b | *P. brassicae* | CBS 178.87 | *Brassica* sp. | Germany | – |
| 8d | *P. syringae* | 947 | *Malus domestica* | SH, China | – |
| | *P. syringae* | CBS 132.23 | *Malus domestica* | UK | – |
| 1 | *P. cactorum* | Pcac2 | *Rosa chinensis* | FJ, China | – |
| | *P. infestans* | Pin1 | *Solanum tuberosum* | FJ, China | – |
| | *P. nicotianae* | Pni1 | *Nicotiana tabacum* | YN, China | – |
| | *P. tentaculata* | Pte1 | *Saussurea costus* | YN, China | – |
| 2 | *P. capsici* | Pcap1 | *Capsicum annuum* | JS, China | – |
| | *P. citrophthora* | Pcitro1 | *Citrus reticulata* | JS, China | – |
| 4 | *P. palmivora* | Ppa1 | *Iridaceae* sp. | YN, China | – |
| | *P. quercina* | CBS 789.95 | Rhizosphere of *Quercus cerris* | Germany | – |
| 6 | *P. megasperma* | CBS 305.36 | *Matthiola incana* | CA, USA | – |
| 7 | *P. cinnamomi* | Pcin1 | *Cedrus deodara* | JS, China | – |
| | *P. melonis* | Pme1 (PMNJHG1) | *Cucumis sativus* | JS, China | – |
| | *P. rubi* | CBS 967.95 | *Rubus idaeus* | Scotland, UK | – |
| | *P. sojae* | P6497 | *Glycine max* | MS, USA | – |
| | *P. sojae* | Peng-R3 | *Glycine max* | China | – |
| | *P. sojae* | Peng-R20 | *Glycine max* | China | – |
| | *P. ×cambivora* | CBS 248.60 | *Castanea sativa* | France | – |
| | *P. ×cambivora* | Pcam1 | *Malus domestica* | SH, China | – |
| 10 | *P. boehmeriae* | Pbo1 | *Gossypium* sp. | JS, China | – |
| *Globisporangium* | *Globisporangium ultimum* | Gu1 | Irrigation water | JS, China | – |
| Fungi | *Alternaria alternata* | Aal1 | Soil | JS, China | – |
| | *Botrytis cinerea* | Bci1 | *Cucumis sativus* | JS, China | – |
| | *Bremia lactucae* | Bla1 | *Lactuca sativa* | JS, China | – |
| | *Colletotrichum glycines* | Cgl1 | *Glycine max* | JS, China | – |
| | *C. orbiculare* | Cor1 | *Citrullus lanatus* | JS, China | – |
| | *C. truncatum* | Ctr1 | *Glycine max* | JS, China | – |
| | *Endothia parasitica* | Epa1 | *Castanea mollissima* | JS, China | – |
| | *Fusarium equiseti* | Feq1 | *Gossypium* sp. | JS, China | – |
| | *F. oxysporium* | Fox1 | *Gossypium* sp. | JS, China | – |
| | *F. solani* | Fso1 | *Gossypium* sp. | JS, China | – |
| | *F. solani* | Fso2 | *Glycine max* | JS, China | – |
| | *Magnaporthe grisea* | Guy11 | *Oryza sativa* | French Guiana | – |

| (Sub)clade[a] | Species | Isolate | Host/substrate | Location[b] | RPA–LFD[c] |
|---|---|---|---|---|---|
| | *Rhizoctonia solani* | Rso1 | *Gossypium* sp. | JS, China | – |
| | *Tilletia indica* | Tin1 | *Triticum aestivum* | JS, China | – |
| | *Verticillium dahliae* | Vda1 | *Gossypium* sp. | JS, China | – |

Note:
[a] Molecular phylogenetic (sub)clade of *Phytophthora* species as indicated by *Martin, Blair & Coffey (2014)* and *Yang, Tyler & Hong (2017)*.
[b] Abbreviations of provinces and municipality in China: SH, Shanghai; JS, Jiangsu; FJ, Fujian; YN, Yunnan.
[c] Positive (+) or negative (−) detection result in the RPA–LFD assay for detecting *P. hibernalis*.

**Table 2 Oligonucleotide primers and probe designed for the recombinase polymerase amplification-lateral flow dipstick assay.**

| Name | Sequence (5′–3′) | Length (mer) | GC-content (%) |
|---|---|---|---|
| PhRPA-F primer | TTCCACCCTTCCACCAGACTGCTGAGGAGG | 30 | 60.0 |
| PhRPA-R primer | Biotin-TGTTAGCTGCGTGTTCGTTGGTCACCCCAGA | 31 | 54.8 |
| PhProbe | FAM-CTTCTGTGATTTATCCAGAAAATCCGTACGAT-THF-GAGCTGGACGGCAAGA-C3 space | 48 | 45.8 |

each reaction was then added to the freeze–dried reaction pellet of the TwistAmp® nfo Kit. DNA template (2 μL) and 2.5 μL of magnesium acetate (280 mM) were added into the mixture. Reactions were carried out at 38 °C. To determine the optimal amplification duration, reactions were performed for 5, 10, 15, 20, 25, and 30 min. Each duration was tested three times. Each reaction included 2 μL of the gDNA (10 ng per μL) of *P. hibernalis* isolate 9099. After amplification, 10 μL of RPA product was mixed with 90 μL of phosphate buffered saline with Tween 20 (PBST) running buffer then 10 μL of RPA product-PBST mixture was pipetted to the sample pad of a HybriDetect one LFD (Milenia Biotec, Giessen, Germany). The dipstick was dipped into a tube containing 200 μL of PBST for up to 5 min until a clear control line was visible. This visualization step using LFD was performed at room temperature (average 22 °C). When test and control lines were simultaneously visible, the detection result was positive. If only control line was visible, it indicated a negative detection result. After visualization, LFDs were air-dried and photographed using a Canon PowerShot SX730 HS camera.

## Sensitivity of RPA–LFD and PCR assays

Ten-fold dilutions of gDNA of *P. hibernalis* isolate 9099 including 100, 10, 1, 0.1, 0.01, 0.001 and 0.0001 ng per μL were used as templates in the RPA–LFD assay (20 min RPA). nfH₂O was used in place of the DNA template as a no-template control (NTC) included in each set of reactions. This evaluation of sensitivity was performed three times.

To compare the sensitivity of the RPA–LFD assay to conventional PCR, the same 10-fold dilutions of gDNA of isolate 9099 were used in a PCR assay. Each dilution was used as the DNA template (2 μL) in each 50-μL PCR reaction. Each reaction also included 25 μL of 2× Taq Master Mix (containing Taq DNA Polymerase, dNTP and an optimized buffer; Vazyme Biotech, Nanjing, China), 2 μL of each of PhRPA-F and PhRPA-R primers (10 μM) and 19 μL of nfH₂O. PCR reactions were completed in a SimpliAmp™ thermal

cycler instrument (Model A24812, Thermo Fisher Scientific, Wilmington, DE, USA) following an initial denaturation step at 95 °C for 3 min, 35 cycles of 95 °C for 15 s, 56 °C (optimized) for 15 s, and 72 °C for 15 s, plus a final extension at 72 °C for 5 min. Each set of PCR reactions included an NTC. PCR products were examined in 1% agarose gel electrophoresis at 120 V for approximately 25 min. Agarose gel was stained by ethidium bromide and visualized on a transilluminator. The PCR assay was carried out three times.

### Specificity of the RPA–LFD assay

The RPA–LFD assay was evaluated using gDNAs of 45 isolates of 39 species (Table 1). Each reaction included 2 μL of DNA template (10 ng per μL). The assay was performed three times against each isolate.

### Detection of *P. hibernalis* in artificially inoculated orange fruits using RPA–LFD

Prior to inoculation, orange fruits (*Citrus sinensis*) were washed in distilled water, immersed in 70% ethanol for 10 sec and then rinsed with distilled water. Each fruit was wounded (approximately 2 cm depth) using a sterile inoculation needle. A 5-day-old cV8A plug (1 × 1 cm) of *P. hibernalis* isolate 9099 was placed onto the wound site of each of six replicate fruits and secured with parafilm. A sterile cV8A plug was used on a non-inoculated control fruit. All fruits were then placed on two layers of wet filter papers in a container and stored in a dark incubator set at 20 °C. After four days, symptoms resembling those of brown rot were observed around the wound sites of inoculated fruits. Fruits were rinsed with distilled water to remove the remaining agar plugs. Both flesh and peel tissues around the wound site were cut into approximately 1 × 1 cm pieces and placed onto PARP-cV8A selective media (*Jeffers & Martin, 1986*) to recover *Phytophthora* isolates at 20 °C. Total DNAs of these tissues were extracted using a previously described NaOH method (*Wang, Qi & Cutler, 1993*). Briefly, 20 mg of plant tissues collected from the wound site of each fruit were placed into a 1.5 mL microtube containing 200 μL of NaOH (0.5 N). They were grinded for approximately 1 min until no large pieces of plant tissues were visible using a sterile tissue grinder pestle. Then, 5 μL of grinded tissues in NaOH were transferred to a new microtube containing 495 μL of Tris buffer (100 mM, pH 8.0). Two μL of the mixture were used as the DNA template in each RPA reaction. Isolate 9099 gDNA (10 ng per μL) and nfH$_2$O were used as a positive control and NTC, respectively. This experiment was repeated once.

## RESULTS

### Optimal duration of RPA

RPA was performed using isolate 9099 gDNA (10 ng per μL) for different amplification durations ranging from 5 to 30 min with 5-min intervals. Both test and control lines were visible on all dipsticks in three repeats of the experiment. Whereas the test lines correlated to 5 min of RPA were weak, as the amplification extended to 10 min and longer, they increased in intensity (Fig. 1). Based on the finding, the amplification duration was set at 20 min for the subsequent RPA reactions.

**Duration (min) of RPA**

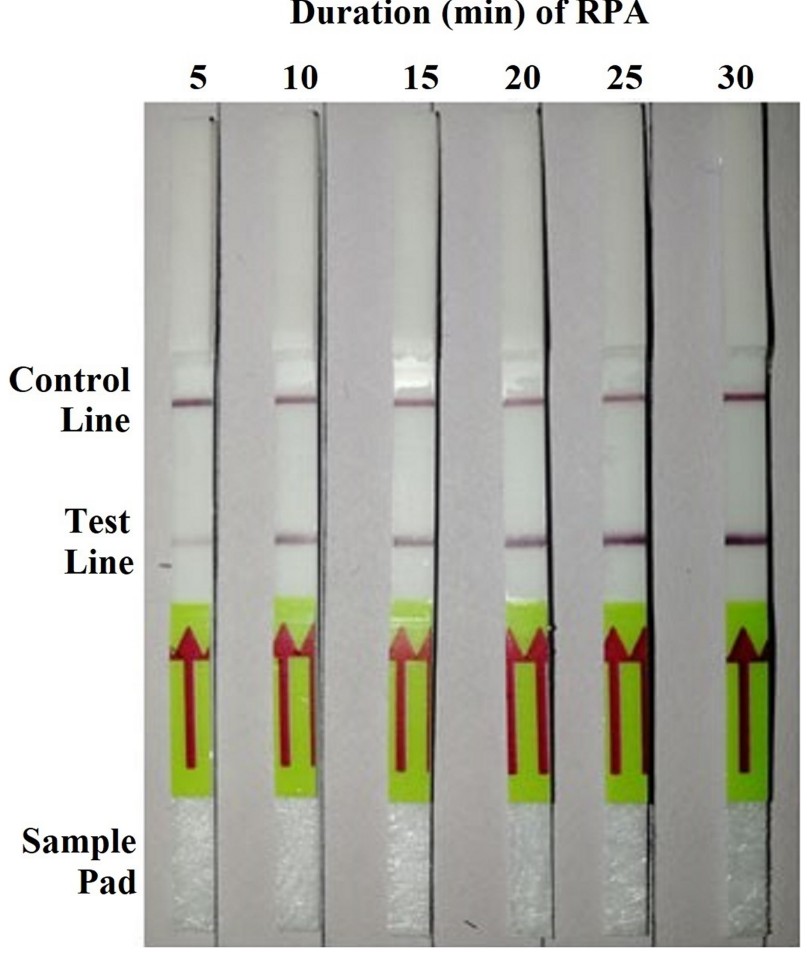

**Figure 1** Detection of the genomic DNA (10 ng per μL) of *Phytophthora hibernalis* isolate 9099 using the recombinase polymerase amplification-lateral flow dipstick (RPA–LFD) assay with various amplification durations ranging from 5 to 30 min with 5-min interval.

## Sensitivity and specificity of the RPA–LFD assay

In the sensitivity evaluation of the RPA–LFD assay, all dipsticks yielded visible control lines, indicating valid tests. Test lines were visible on all dipsticks correlating to RPA reactions using ≥0.2 ng (2 μL, 0.1 ng per μL) gDNA of *P. hibernalis* isolate 9099 (Fig. 2A). No test lines were observed on dipsticks using ≤0.02 ng of gDNA (Fig. 2A). In the comparative evaluation of the PCR assay, ≥2 ng of gDNA was required in each PCR reaction for the successful amplification using primers PhRPA-F and PhRPA-R (Fig. 2B). The above results were consistent among three repeats of each assay.

In the specificity evaluation of the RPA–LFD assay, all LFDs had visible control lines. Only the reactions using gDNAs *of P. hibernalis* isolates yielded positive detection results (Table 1). The results were consistent among three repeats of the experiment against each isolate.

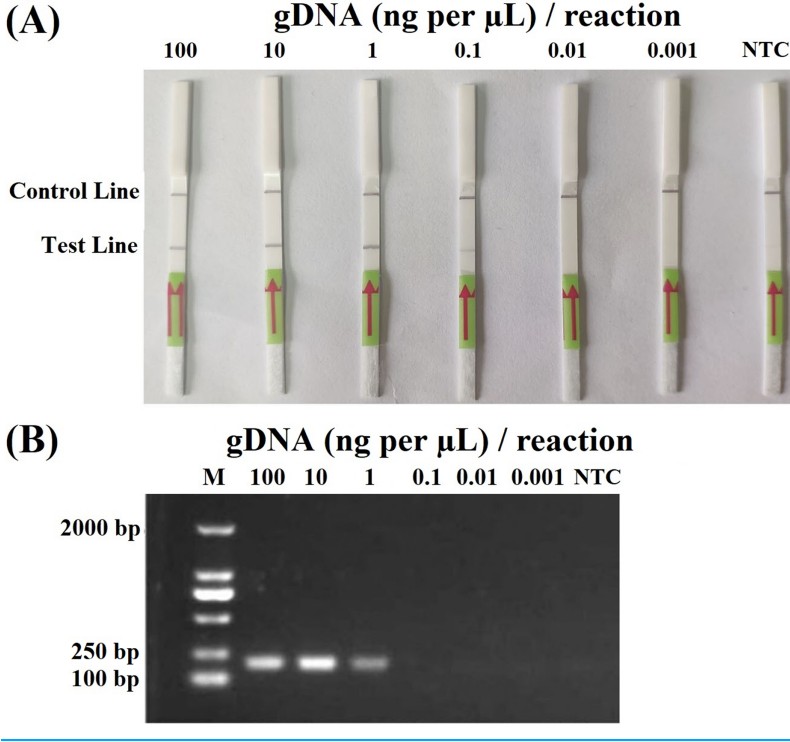

**Figure 2 Sensitivity of the recombinase polymerase amplification-lateral flow dipstick (A) and PCR (B) assays using 10-fold dilutions of genomic DNA (gDNA; ng per μL) of *P. hibernalis* isolate 9099.** Two μL of gDNA were added to each 50-μL reaction. Nuclease-free water was used in no-template controls (NTC). M: NormalRun$^{TM}$ prestained 250bp-I DNA ladder (GsDL2501; Generay Biotech, Shanghai, China).

## Detection of *P. hibernalis* in artificially inoculated orange fruits using RPA–LFD

In the first repeat of the experiment, *P. hibernalis* isolates were recovered from all six inoculated orange fruits, which had shown typical brown rot symptoms at the wounded sites four days after inoculation, while not from the non-inoculated control fruit. In the RPA–LFD assay, *P. hibernalis* DNAs were detected in total DNAs extracted from six artificially inoculated fruits, while the non-inoculated control lacked detection of *P. hibernalis* (Fig. 3). In the second repeat experiment, only three of six inoculated fruits had shown brown rot symptoms after four days, while the other three as well as the non-inoculated control remained asymptomatic. Correspondingly, total DNAs of three symptomatic fruits yielded positive detection of *P. hibernalis* in the RPA–LFD assay, whereas the remaining fruits gave rise to negative results (Fig. 3).

## DISCUSSION

A novel RPA–LFD assay for detecting *P. hibernalis*, the quarantine citrus pathogen in China, was developed in this study. This assay consistently detected 0.2 ng of *P. hibernalis* genomic DNA in a 50-μL RPA reaction (4 pg per μL in the reaction). It also was demonstrated as specific to *P. hibernalis*, while yielded no detection against 43 isolates belonging to 38 non-*P. hibernalis* species. Furthermore, this assay was able to detect

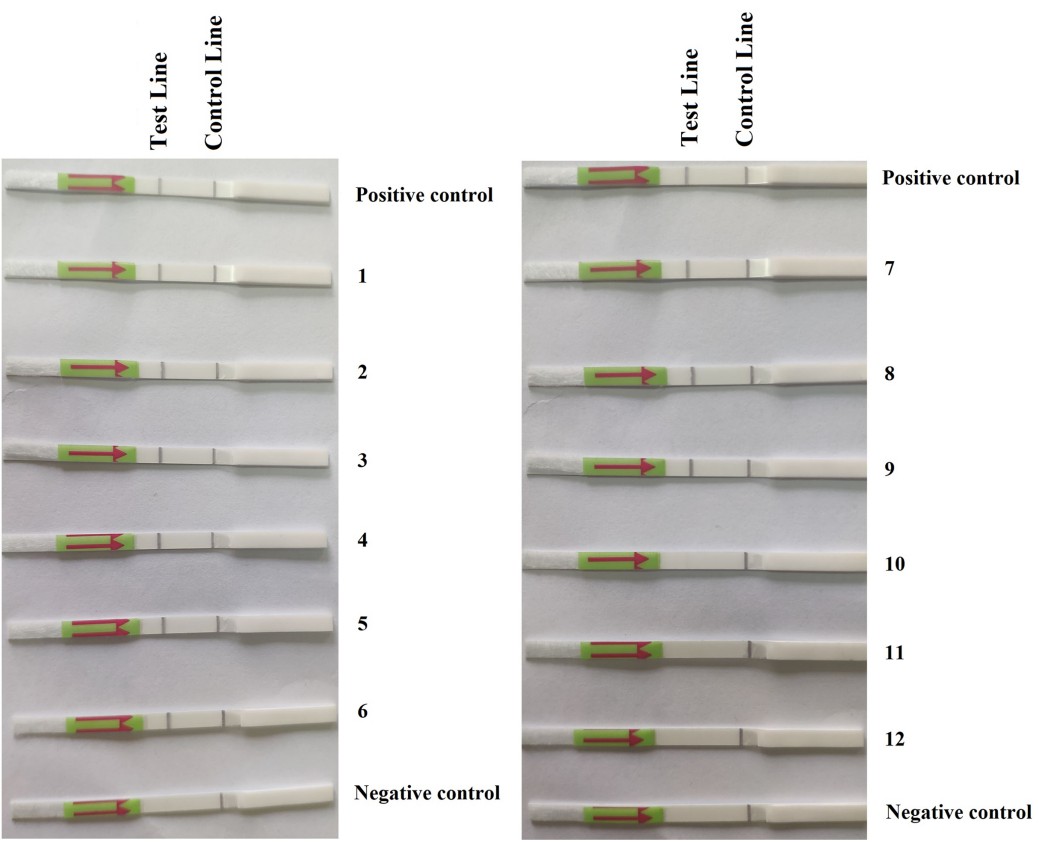

**Figure 3 Detection of *P. hibernalis* in artificially inoculated orange fruits using the recombinase polymerase amplification-lateral flow dipstick assay.** Positive control: genomic DNA (10 ng per μL) of *P. hibernalis* isolate 9099; 1–6: inoculated, symptomatic fruits in the first repeat experiment; 7–9: inoculated, symptomatic fruits in the second repeat; 10–12: inoculated, asymptomatic fruits in the second repeat; Negative control: non-inoculated fruits.  

*P. hibernalis* from artificially inoculated orange fruits, indicating its potential application on field samples.

Compared to PCR-based methods, this RPA–LFD assay has several advantages. First, it requires much shorter time span of 15–35 min, including 10–30 min for the RPA step (Fig. 1) and 5 min for the visualization of detection results using LFDs, whereas a typical 30-cycles PCR requires approximately 90 min. Second, RPA in this study was carried out at a constant temperature of 38 °C, while this amplification step theoretically could be done within a wider temperature range of 25–45 °C (*Daher et al., 2016*; *James & Macdonald, 2015*). The visualization step using LFDs was performed at room temperature (average 22 °C). In contrast, PCR-based methods require thermocyclers for stringent temperature control. Third, the results of RPA could be visualized on LFDs promptly with naked eyes (Fig. 2A), while examining PCR results typically required gel electrophoresis and fluorometers (Fig. 2B).

This novel RPA–LFD assay has great potentials of use for specific detection of *P. hibernalis* on citrus fruits. As demonstrated in this study, this assay was able to detect *P. hibernalis* using total DNA of *P. hibernalis*-infected fruits (Fig. 3), while lacked

**Table 3 Comparison of estimated costs of the recombinase polymerase amplification-lateral flow dipstick (RPA–LFD) assay and conventional PCR for processing 10 samples.**

| Material | Cost (US $)/unit | RPA–LFD | | PCR | |
|---|---|---|---|---|---|
| | | Amount | Cost | Amount | Cost |
| Labor | 15/h[a] | 0.5 | 7.5 | 3 | 45 |
| Gloves | 0.1/pair | 1 | 0.1 | 1 | 0.1 |
| Pipette tips (0–200 μL) | 0.03 each | 10 | 0.3 | 10 | 0.3 |
| Pipette tips (0–10 μL) | 0.03 each | 10 | 0.3 | 30 | 0.9 |
| Microtubes 1.5 mL | 0.02 each | 22 | 0.44 | 1 | 0.02 |
| Microtubes 0.2 mL | 0.03 each | n/a[b] | n/a | 10 | 0.3 |
| TwistAmp® nfo reaction | 3.13 each[c] | 10 | 31.3 | n/a | n/a |
| HybriDetect 1 Lateral Flow Dipstick | 2.75 each[d] | 10 | 27.5 | n/a | n/a |
| 2× Taq Master Mix | 0.006/μL | n/a | n/a | 250 | 1.5 |
| Forward oligonucleotide (10 μM) | 0.15/μL | 21 | 3.15 | 20 | 3 |
| Reverse oligonucleotide (10 μM) | 0.15/μL | 21 | 3.15 | 20 | 3 |
| Oligonucleotide probe (10 μM) | 0.5/μL | 6 | 3 | n/a | n/a |
| Electrophoresis | 0.6/sample | n/a | n/a | 10 | 6 |
| Gel staining and imaging | 0.2/sample | n/a | n/a | 10 | 2 |
| Losses (10%)[e] | | | 7.67 | | 6.21 |
| Indirect costs[f] | | | 3 | | 18 |
| Total for 10 samples | | | 87.41 | | 86.33 |

**Note:**
[a] Typical hourly salary of skilled biological technicians at the plant quarantine departments of customs in Jiangsu and Shanghai, China.
[b] n/a = not applicable.
[c] Each TwistAmp® nfo Kit (£240) includes 96 reactions.
[d] Milenia HybriDetect one (pack of 100) costs £220.
[e] Ten percent on the total amount of direct costs was defined a priori, the losses of consumables that occur during the execution of techniques (*Schlatter et al., 2015*).
[f] Approximate indirect costs of facility security and maintenance, purchase and maintenance of special equipment such as thermal cycler, gel electrophoresis system, and transilluminator, and electricity.

detection against any other tested species (Table 1) including four other important *Phytophthora* species that are pathogens of citrus crops, namely *P. citrophthora*, *P. nicotianae*, *P. palmivora* and *P. syringae*. Such specificity in conjunction with its high sensitivity make this assay an ideal primary screening tool for detecting *P. hibernalis* in diagnostic laboratories and plant quarantine departments at customs.

In addition to high specificity and sensitivity, field assays must have short time span and be possible to perform without special equipment (*Yu et al., 2019*). The RPA–LFD assay developed in this study could be performed within 25 min. We also implemented a modified NaOH method for extracting RPA-ready DNAs from *P. hibernalis*-infected fruits within 5 min. Such short time span and low requirement for special equipment and operation skills allow first responders and diagnosticians process a large quantity of samples under time- and resource-limited conditions. Different rapid DNA extraction methods have been used for other RPA assays detecting *Phytophthora* species. For example, *Miles, Martin & Coffey (2015)* tested six crude extraction buffers using *Fragaria × ananassa* crown tissues and *P. cactorum* DNA and subsequently chose the

GEB2 buffer (ACC 00130, Agdia, Elkhart, IN, USA) to rapidly prepare DNA samples for a *Phytophthora* genus-specific RPA assay. Yu et al. (2019) used a cellulose dipstick method to obtain amplification-ready DNA for an RPA–LFD assay detecting *P. capsici*. It will be meaningful to evaluate these extraction media against various host tissues and compare their influence on the efficacy of RPA assays in future studies.

While saving time, the RPA–LFD assay is comparable to PCR in cost-effectiveness. Estimated costs for processing 10 samples at customs in Jiangsu and Shanghai, China are similar between RPA–LFD and PCR (Table 3). The RPA–LFD assay has the advantage in reducing labor costs and avoiding both direct and indirect costs associated with special equipment such as thermal cycler and gel electrophoresis system (Table 3). While the materials of RPA–LFD including the TwistAmp® nfo reaction reagents and HybriDetect one LFDs are currently more costly than those of PCR (Table 3), shorter diagnostic time at the customs reduces berthing fees for shipping companies, conserves important shelf life of fresh products such as fruits and vegetables and subsequently reduces prices for customers. The exact cost-effectiveness of RPA assays for the diagnosis of quarantine plant pathogens warrants more detailed evaluations.

## ACKNOWLEDGEMENTS

Mention of trade names or commercial products in this publication is solely for the purpose of providing specific information and does not imply recommendation or endorsement by the U.S. Department of Agriculture. USDA is an equal opportunity provider and employer.

### Funding

This work was supported by the National Natural Science Foundation of China (31500526), China Postdoctoral Science Foundation (2016T790467), Overseas Research and Study Project of Excellent Young and Middle-aged Teachers and Principals in Colleges and Universities of Jiangsu Province of 2018, the Priority Academic Program Development of Jiangsu Higher Education Institutions, Special Fund for Agro-scientific Research in the Public Interest (201503112), Jiangsu Basic Research Program (Natural Science Foundation) Project (BK 20191389) and Jiangsu University Natural Science Research Surface Project (19KJB220003). The funders had no role in study design, data collection and analysis, decision to publish, or preparation of the manuscript.

### Grant Disclosures

The following grant information was disclosed by the authors:
The National Natural Science Foundation of China: 31500526.
China Postdoctoral Science Foundation: 2016T790467.
Overseas Research and Study Project of Excellent Young and Middle-aged Teachers and Principals in Colleges and Universities of Jiangsu Province of 2018, the Priority Academic Program Development of Jiangsu Higher Education Institutions.

Special Fund for Agro-scientific Research in the Public Interest: 201503112.
Jiangsu Basic Research Program (Natural Science Foundation) Project: BK 20191389.
Jiangsu University Natural Science Research Surface Project: 19KJB220003.

## Competing Interests

Binbin Jiao is employed by the Technical Center for Animal, Plant and Food Inspection and Quarantine of Shanghai Customs. Wen Tian is employed by Jiangyin Customs House. The authors declare that they have no competing interests.

## Author Contributions

- Tingting Dai conceived and designed the experiments, performed the experiments, analyzed the data, contributed reagents/materials/analysis tools, prepared figures and/or tables, authored or reviewed drafts of the paper, approved the final draft.
- Tao Hu performed the experiments, analyzed the data, authored or reviewed drafts of the paper, approved the final draft.
- Xiao Yang conceived and designed the experiments, analyzed the data, prepared figures and/or tables, authored or reviewed drafts of the paper, approved the final draft.
- Danyu Shen conceived and designed the experiments, analyzed the data, contributed reagents/materials/analysis tools, authored or reviewed drafts of the paper, approved the final draft.
- Binbin Jiao analyzed the data, contributed reagents/materials/analysis tools, authored or reviewed drafts of the paper, approved the final draft.
- Wen Tian analyzed the data, prepared figures and/or tables, authored or reviewed drafts of the paper, approved the final draft.
- Yue Xu performed the experiments, analyzed the data, authored or reviewed drafts of the paper, approved the final draft.

## Data Availability

The raw data is available in the Supplemental Files.

## Supplemental Information

Supplemental information for this article can be found online at http://dx.doi.org/10.7717/peerj.8083#supplemental-information.

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
