# Peer review of "A recombinase polymerase amplification-lateral flow dipstick assay for rapid detection of the quarantine citrus pathogen in China, Phytophthora hibernalis"

_PeerJ, doi:10.7717/peerj.8083_

## Round 0.1 · original submission · Major Revisions

Based on the reviews conducted by two major reviewers and my own scientific assessment, your manuscript still needs a Major revision before it can be considered for publication in PeerJ

·

Basic reporting

The writing is clear and although I am not a native english speaker, it seems to me that professional English is used.
The littererature references are adequately quoted, although some of them should be discussed and compared to the results obtained in this study. In particular, the advantages/drawbacks of the tests already available in the literature should be discussed, and not only from the turnaround time for analysis perspective.
The structure of the article is adequate, and the figures are informative and of very good quality. An additional figure showing interspecific polymorphisms within Ypt gene would be welcome though (see general comments).

Experimental design

The authors do not justify the choice of Ypt gene as a target for this new species-specific test. This should be indicated.
The way or rationale that primers and probes have been designed and the final set selected is not sufficiently described. It means that the selection of the primers and probes would not be easy to replicate by others.
Although the RPA technique was developed by other research teams, as quoted in the manuscript, it would be good for the reader to get a reminder of how it actually works. Most of the readers, including myself, are not familiar with this technique and it would be interesting to have more information about that, rather than quote article(s) to refer to.

As we are dealing here with a quarantine pathogen, the specificity and sensitivity of the test is of paramount importance. I believe that the specificity of the test has been unsufficiently addressed here, and verification of this performance criteria should be improved with a larger and more relevant set of fungal strains (see general comments). Also it would have been interesting to check whether the specificity (not only the sensitivity) remains optimal with disturbed parameters (i.e. duration increased, temperature decrease, etc.), as may occur during routine use.

Validity of the findings

Raw data about the selection of primers and probe should be made available (sequence alignmements, number of sequences and strains used, etc.).

Additional comments

Although I do not question at all the validity of the finding, i.e. a set a primers and probe that enable amplification of P. hibernalis DNA, I feel that the current version of the manuscript only contains a limited amount of information regarding the selection and design of the primers and probe, and insufficient data regarding performance criteria.
Although RPA is not a widely used technique, it looks very promising indeed. The reader may like to get more information about how it works. The experimental process used to design and select the primers and probe is not described enough here. One should have more information about the selection of the YPT1 gene among other genes, which types of polymorphisms were targeted by the oligonucleotides, etc. A shortcoming of this study is the lack of information about the selection of species-specific oligonucleotides. The reader should know how many sequences were used for alignment of Ypt gene sequences, how many and which species were represented.
The meaning or role of the different features of the oligonucleotides (e.g. labels, spacer, base analog) should be described. In addition, what is the role of the labeled probe, since there is apparently no fluorescence monitoring during the run? Does it bring an additional level of specificity or is it only used for amplification of part of the gene (as can be read L79-80)?
There are two DNA amplification kits reported (‘TwistAmp DNA amplification kits’ and ‘TwistAmp nfo kit’). It is unclear why two different kits were used, at different steps of the selection and use of the primers and probe.
Regarding the wet lab validation of specificity, a quite large range of Phytophthora species were assessed, but one may regret that few Clade 8 Phytophthora species were included (i.e. P. lateralis, P. foliorum). Although they are unlikely to be found on citrus, they are genetically very close, and the specificity of the test should be demonstrated with closely related species. In the same vein, fungal species frequently found on citrus fruit should have been included in the specificity assessment. Why including only genera and species that are not citrus pathogens or endophytes? It would be good to include species causing lesions on citrus fruits or commonly found as endophytes and check that their DNA does not yield cross-reactions and false-positive results.
Also it would have been interesting to check whether the specificity (not only the sensitivity as reported) remains optimal with disturbed parameters (i.e. duration increased, temperature decrease, etc.), as it may occur during routine use. One may imagine that less stringent parameters might unfortunately enable a weak amplification of non-target DNA, and in consequence, a faint band on the strip. A test targeting a quarantine pathogen deserves careful validation.

The protocol followed to extract DNA from fruit lesion should be a bit more detailed. One should not have to go back to another article (Wang et al.) to have the practical information. In addition, in the manuscript (L146) it is not clear what kind of fruit tissue are used for DNA extraction (the ones that were plated before?), and how they were crushed.

The assay developed in this work is aimed at being used "to detect the quarantine pathogen P. hibernalis in diagnostic laboratories, plant quarantine departments, and customs, especially under time- and resource limited conditions. " RPA technique probably meets these requirements, but there are no data in the paper that demonstrate its cost-effectiveness and robustness for instance. This would be good to discuss.
Last, the discussion section looks a bit meager, and would benefit from comparison with other P. hibernalis detection tools available in the literature. A critical review of potential pitfalls of the different techniques (including RPA of course) would be appreciated.

Reviewer 2 ·

Basic reporting

The manuscript is written in clear English, structurally conformed to the format of PeerJ, and self-contained.
However, the authors need provide sufficient field background/context:
1. Line 38 to 41: The authors should have an accurate description of P. hibernalis: the disease and host range.
2. Line 62 to 63: The authors need to have sufficient background information on the lateral flow assays in the “Introduction”: the principle of the lateral flow immunochromatographic assays; why and how does the LFD accurately detect the target amplicon from the RPA; can the LFD be used to quantify/semi-quantify the target amplicon from the RPA? This is very important for the readers to understand the methodology of this manuscript.
The tables and figures are relevant to the content. My main concern is:
3. Fig. 3: The data shown in Fig. 3 are all included in Table 2. So I suggest the authors to only keep Table 2 in the manuscript.
I have some suggestions on the content and labeling of Table 1 and Fig. 2.

Experimental design

The authors clearly explained the reason and advantages to developing a RPA assay for P. hibernalis detection, however, as I mentioned before, they need to provide sufficient background information on the lateral flow assays to justify the utilization of LFD test.
In “M&M”, some more details, information, and explanations need to be added/addressed:
4. Line 80: I don't understand why a species probe was designed and how it was used in the RPA-LFD assay. Please explain.
5. Line 134: The authors tested the specificity of their RPA-LFD assay against other Oomycota and fungal species, however, they did not include Phytophthora citrophthora, which is an important citrus pathogen (Erwin & Ribeiro 1996; Graham & Menge 2000). Please explain why you did not include P. citrophthora isolates.
6. Line 137 to 146: About the inoculation of orange fruit by P. hibernalis agar culture plugs, did you put the plugs (1cm x 1cm) into the wound (made by a needle) or on the wound? How to secure the plugs if you put it on the wound? According to Erwin & Ribeiro (1996), the maximum growth temperature of P. hibernalis is <25C, I’m wondering what symptoms you observed after 5 days at 25C. What tissue type did you sample? (peel or flesh?). I assume the agar culture plug was removed when sampling? Or it was dried out? I think using a zoospore suspension of Phytophthora spp. may be a more preferable method for fruit inoculation, because it’s easy to quantify and more comparable to the infection occurring in nature.

Validity of the findings

In “Results” and “Discussion” parts, the authors described/explained their data from the experiments, and stated their conclusions. I think there are a few major gaps/concerns that need to be addressed.
7. Line 175: What were the typical brown rot symptoms you observed on the fruit 5 days after inoculation at 25C? P. hibernalis has lower optimal growth and infection temperatures (<20C).
8. Line 176 to 178: In Fig. 4, P. hibernalis mycelial DNA 20 ng was used as the positive control. The total DNA of the infected orange fruit tissue used for RPA-LFD was also 20 ng as described in the M&M. The total DNAs of the infected plant tissue should contain a large portion of plant DNA, which makes the pathogen DNA much less than 20 ng when loading 20 ng total DNA. Please explain why the intensity of test lines of the symptomatic fruit samples were as intense as the positive control.
9. Line 199 to 200: To validate this conclusion, the authors need to 1) explain how the RPA-LFD assays quantify the target sequences; 2) discuss the validation/optimization PCR reactions using the newly designed primers
10. Line 189, 203 to 205: The authors stated that the RPA-LFD assay has great potential of use for specific detection of P. hibernalis on field samples/citrus fruit. However, they only tested the assays with inoculated orange fruit tissues. I think they should have tested the assay on some other field samples. It will be interesting to know if the plant DNA will affect the sensitivity of RPA-LFD assay to detect the pathogen DNA.
11. Line 207 to 211: The PRA assay developed here still requires a sample DNA extraction step before the test, which requires the molecular skills to accomplish. In Miles et al. (2015), they used crude extracts of field samples to validate their Phytophthora genus/species specific RPA assay. I want to know what the author thinks about this method.

Additional comments

Please see above and the attached pdf file for my other comments and suggestions.

Annotated reviews are not available for download in order to protect the identity of reviewers who chose to remain anonymous.

Reviewer 3 ·

Basic reporting

• The paper is clear, and the language used is correct for the target audience.
• All the references are relevant for the purpose of the paper, and I only suggest the authors to consider citing the reference below to put P. hibernalis in the context of the other Phytophthora species to understand the selection of the isolates for specificity and the design of the assay.
o Martin, F. N., Blair, J. E., & Coffey, M. D. (2014). A combined mitochondrial and nuclear multilocus phylogeny of the genus Phytophthora. Fungal Genetics and Biology, 66, 19–32. http://doi.org/10.1016/j.fgb.2014.02.006
• All Figures are relevant and I will suggest to additionally present other information in a supplementary material (see below).

Experimental design

• The experimental design is consequent with target of the paper, the authors have done a good job on following logic steps for developing and validating the isothermal amplification assay for the detection of P. hibernalis.
• The scope and the goal of the paper are pretty clear, and the authors managed to demonstrated the functionality of the assay developed. The only limiting factor will be the use of the test on real samples that could potentially be infected with P. hibernalis.
• Methods are described clearly, and these could be reproduced with detail described by the authors.

Validity of the findings

• All the findings of the paper are valid and provides a novel assay to detect Phytophthora hibernalis using isothermal amplification with minimal requirements, which could be really useful on limited setups and for control of plant material imports.
• All the data has been provided.

Additional comments

The paper by Dai et al. provides a development in diagnosis to scout plant material for P. hibernalis and the authors have done a good job on describing the development process, validation and further use of the test using material inoculated in lab conditions. The papers is well written and it is simple enough and easy to follow for the most part. The paper is very concise, and the results are presented in clear way.

I would like to have screening of real samples, like the original sample that was used as source for the isolate 9099. However, the pathogen is not present in China and that could be a limiting factor. Nonetheless, the assay seems to work on most artificially inoculated samples.

Most of my comments are minor and these could be easily addressed by the authors.

Minor points
• Line 66 – remove “that of”
• Line 70 – 80 Clarify or mention how many primers were tested and those could be added in the supplementary material indicating their performance (e.g. no amplification, weak, etc.)
• Line 81 – 84 and table 1 should list both nfo probe and the fluorescent probe, indicating very clearly were are all the modification including the abasic site or THF site and where are located the fluorophore and quencher.
• Line 145 -clarify if the isolation media was also V8c or did you use a different media. Also, the concentration of the antibiotics should be included.
• Line 190 – 202. When comparing the conventional PCR with RPA-LFD, it should be acknowledged that you are using the primers designed for RPA, which are at least 30 bp and these are not optimized for conventional PCR. So it could affect the performance of those, if you did optimize the primers either testing different annealing temperatures or duration of cycles please mention that. Otherwise, you should discuss or acknowledge that issue.
• Table 2 – I will recommend adding the Clade information for every Phytophthora tested, and sort those isolates by clade, in that way you facilitate the understanding of how some of those isolates were selected since a good number of them are in the Clade 8 were P. hibernalis is classified. I highlighted those in blue as example on the PDF.
• Figure 2 – you should use the actual amount of DNA rather than using the concentrations or include a note either on the legend or the graph itself that suggest that you used 2 µL.

Annotated reviews are not available for download in order to protect the identity of reviewers who chose to remain anonymous.

---

## Round 0.2 · accepted · Accept

Based on the rebuttal information provided by the correspondence author. The authors of the manuscript have clearly addresses all comments and suggestions made by 3 reviewers. In addition, my own assessment as an Academic Editor, I recommend publishing the "Revised Version" of the manuscript in PeerJ.

Dr. Simon Francis Shamoun
Academic Editor, PeerJ